# Systemic Impact of Subgingival Infection Control in Periodontitis Patients with Cardiovascular Disease: A Narrative Review

**DOI:** 10.3390/antibiotics13040359

**Published:** 2024-04-15

**Authors:** Carmen Silvia Caloian, Andreea Ciurea, Marius Negucioiu, Alexandra Roman, Iulia Cristina Micu, Andrei Picoș, Andrada Soancă

**Affiliations:** 1Department of Periodontology, Faculty of Dental Medicine, Iuliu Hațieganu University of Medicine and Pharmacy Cluj-Napoca, Victor Babeș St., No. 15, 400012 Cluj-Napoca, Romania; caloiancarmen@yahoo.com (C.S.C.); andreea_candea@yahoo.com (A.C.); i.cristina.micu@gmail.com (I.C.M.); andrapopovici@gmail.com (A.S.); 2Department of Prosthodontics, Faculty of Dental Medicine, Iuliu Hatieganu University of Medicine and Pharmacy Cluj-Napoca, Clinicilor St., No. 32, 400006 Cluj-Napoca, Romania; sica5319@yahoo.de; 3Department of Prevention in Dental Medicine, Faculty of Dental Medicine, Iuliu Hatieganu University of Medicine and Pharmacy Cluj-Napoca, Avram Iancu St., No. 31, 400083 Cluj-Napoca, Romania

**Keywords:** periodontitis, bacteria, inflammation, cardiovascular disease, scaling

## Abstract

Introduction: Periodontitis, an infectious inflammatory condition, is a key contributor to sustained systemic inflammation, intricately linked to atherosclerotic cardiovascular disease (CVD), the leading cause of death in developed nations. Treating periodontitis with subgingival mechanical instrumentation with or without adjunctive antimicrobials reduces the microbial burden and local inflammation, while also potentially bringing systemic benefits for patients with both periodontitis and CVD. This review examines systemic effects of subgingival instrumentation with or without antimicrobial products in individuals with periodontitis and CVD, and explores intricate pathogenetic interactions between periodontitis and CVD. Material and Methods: English-language databases (PubMed MEDLINE and Cochrane Library) were searched for studies assessing the effects of nonsurgical periodontal therapies in periodontitis patients with or without CVD. Results: While the ability of periodontal therapy to reduce mortality- and morbidity-related outcomes in CVD patients with periodontitis remains uncertain, some studies indicate a decrease in inflammatory markers and blood cell counts. Subgingival mechanical instrumentation delivered over multiple short sessions carries lower risks of adverse effects, particularly systemic inflammation, compared to the full-mouth delivery, making it a preferable option for CVD patients. Conclusions: Subgingival mechanical instrumentation, ideally conducted in a quadrant-based therapeutic approach, to decontaminate periodontal pockets has the potential to reduce both local and systemic inflammation with minimal adverse effects in patients suffering from periodontitis and concurrent CVD.

## 1. Introduction

Periodontitis is a chronic, multifactorial infectious disease that triggers a local dysregulated immune–inflammatory response [1], leading to progressive periodontal destruction, significant oral functional impairments, systemic consequences [2], and increased medical costs [3]. Due to its enormous global burden in the general population, periodontitis has become a major public health problem. It affects around 50% of the population [4], and as of 2019, 1.1 billion severe periodontitis cases have been reported worldwide [5].

Cardiovascular disease (CVD) encompasses various pathologies such as ischemic heart disease, stroke, hypertension, rheumatic heart disease, cardiomyopathies, and atrial fibrillation, with most having an atherosclerotic origin [2]. CVD is the leading cause of mortality globally, responsible for 32% of all deaths [6]. In Europe alone, in 2017, over 3.9 million deaths were attributed to CVD [7].

Periodontitis, mostly stage III and IV phenotypes, through the considerable subgingival bacterial load, can contribute to persistent systemic inflammation, impacting the development of atherosclerotic CVD [8,9,10] and its complications [11]. Epidemiological studies have reported positive associations between periodontal disease and coronary heart disease and stroke [10]. Moreover, an increased risk of developing CVD among periodontitis patients has been identified [12,13,14,15]. Current evidence now recognizes periodontitis as an independent risk factor for atherosclerotic CVD, such as hypertension, coronary artery disease, peripheral artery disease, and their acute events [9,16]. The prerequisite pathogenetic substrate of atherosclerosis is endothelial dysfunction [17]. 

Numerous reviews and meta-analyses based on epidemiological studies have consistently indicated a relationship between periodontitis and CVD with an increased risk of periodontitis patients developing CVD [14,18,19,20,21] or its acute event—myocardial infarction [12,22,23,24,25]. Periodontitis and its major sequela—edentulism—increased the risk of all-cause mortality and mortality due to CVD, coronary heart disease, or cerebrovascular diseases [26]. However, so far, it has not been possible to establish the nature of the association between the two diseases [27].

Excessive and dysbiotic subgingival microbiota as well as aberrant periodontal inflammation are central elements of the pathogenic network inducing local periodontal destruction as well as systemic consequences (Figure 1A). Periodontitis triggers the initiation and progression of endothelial dysfunction, atherosclerosis, and CVD through a networking of pathogenetic mechanisms such as the release of several systemic proinflammatory molecules, unbalanced oxidative stress, the decrease of nitric oxide (NO) concentration [27], upregulation of serum cell-free DNA (cfDNA) levels, dysfunction of endothelial progenitors [28,29,30,31,32,33], or elevated levels of trimethylamine-N-oxide [33,34] induced by periodontitis-related gut dysbiosis [33,35]. 

The implication of periodontopathogens in the development of atheromatous lesions has been suggested by their presence in atheromatous lesions [36]. Identification of periodontal bacterial DNA in coronary atheroma supports their systemic dissemination from periodontal pockets [37]. *Porphyromonas gingivalis* was more frequently detected in coronary atheromatous plaque samples than *Aggregatibacter actinomycetemcomitans* [38]. However, it is difficult to state that DNA fragments belong to living resident bacteria from the atheromas participating in vascular pathology; instead, they may merely be local bystanders [37]. Viable bacteria have been reported to be present in some thrombus samples, all of them identified as positive for bacterial DNA through qPCR assay [39]. 

While there were high hopes that reducing serum inflammatory markers through periodontal therapy would provide preventive benefits for CVD patients, uncertain evidence has accumulated regarding the systemic benefits provided by reducing the microbial burden through subgingival mechanical instrumentation (scaling) in patients with chronic periodontitis [40,41]. 

The present review aims to provide a comprehensive overview of the systemic effects induced by nonsurgical periodontal treatment, consisting of subgingival mechanical instrumentation with or without adjunctive antimicrobials, in patients suffering from both periodontitis and CVD. Additionally, this review focuses on the short-term adverse effects of mechanical instrumentation and on the complex links sustaining the pathogenetic interactions between periodontitis and CVD.

## 2. Results

A total number of 27 full-text papers were identified, from which 19 (7 reviews and 12 clinical trials) reporting on systemic effects of nonsurgical periodontitis treatment in patients with both CVD and periodontitis and 8 (2 reviews and 6 clinical trials) on short-term adverse effects of nonsurgical periodontitis treatment.

### 2.1. Medium-Term Effects of Subgingival Mechanical Instrumentation on Mortality- and Morbidity-Related Outcomes in Cardiovascular Disease Patients with Periodontitis

Several studies have documented the systemic impact of subgingival mechanical instrumentation, which is considered the gold-standard therapy for periodontitis [42], both with and without antimicrobial agents, in patients with CVD. The effects are depicted schematically in Figure 1B.

The most recent systematic review on this subject [40] analyzed the results of two RCTs from the perspective of fatal and nonfatal cardiovascular events. One parallel-arm, double-blind RCT on primary prevention of CVD in a group of periodontitis and metabolic syndrome patients (165 participants) [43] reported one death due to fatal myocardial infarction in the intervention group (subgingival mechanical instrumentation plus double antibiotic regimen) and another three nonfatal events; it could not be identified whether intervention would have reduced the incidence of all-cause death [Peto odds ratio (OR) 7.48, 95% confidence interval (CI) 0.15 to 376.98] or all CVD-related death (Peto OR 7.48, 95% CI 0.15 to 376.98) as compared with supragingival instrumentation alone plus placebo tablets (non-intervention).

Moreover, the subgingival instrumentation combined with an amoxicillin and metronidazole regimen could have increased the risk of cardiovascular events (Peto OR 7.77, 95% CI 1.07 to 56.1) when compared with supragingival instrumentation at the 12-month follow-up [43].

The second study—PAVE (2008)—was a multicenter, parallel-group, single-blind RCT focused on secondary prevention of CVD, which included 303 patients randomly treated with subgingival mechanical instrumentation and oral hygiene instruction (intervention) or oral hygiene instruction plus community follow-up (non-intervention). Unfortunately, only a small group of participants was available for one-year follow-up, which precluded any conclusion on the effects of periodontal treatment on secondary prevention of CVD in terms of all-cause death and all CVD-related death [44,45,46].

### 2.2. Medium-Term Effects of Subgingival Mechanical Instrumentation on Inflammatory Markers and Blood Cell Counts in Cardiovascular Disease Patients with Periodontitis

Clinical studies [33,47,48,49,50] and systematic reviews [51,52,53,54,55] have reported the systemic effects of periodontitis infection control in CVD patients in terms of changes in periodontal parameters or systemic inflammatory markers. Three months after subgingival nonsurgical periodontal therapy, a decrease in serum levels of IL-6 and CRP was observed as compared with a control group receiving only oral hygiene advice [33,47]. More recent information confirms the reduction in serum CRP levels induced by subgingival nonsurgical periodontal treatment [48,49]. Reports from a meta-analysis showed that periodontitis therapy lowered systemic inflammation via reductions in both high-sensitivity CRP (hs-CRP) [0.56 mg/L, 95% confidence interval (CI) (−0.88, −0.25), *p* < 0.001] and IL-6 [0.48 pg/mL, 95% CI (−0.88, −0.08), *p* = 0.020] levels [52]. On the other hand, another meta-analysis [40] reported an uncertain effect of subgingival mechanical instrumentation plus a double antibiotic regimen in lowering serum CRP levels (X coefficient −0.002, 95% CI −0.19 to 0.20) or fibrinogen (X coefficient 10.9, 95% CI −12.0 to 33.8). These uncertainties were accentuated by some reported biases of this meta-analysis [40] related to the methodologies of the included RCTs and some interpretations of the systematic review. Furthermore, the authors [40] failed to account for outdated measures of chronic periodontitis used in the analyzed RCTs [43,44,45,46], and the authors [40] changed its case definition during the review process, which did not correspond either to the one provided by the RCTs or to the current one [56,57].

Other systematic reviews exploring the systemic effects of subgingival infection control in periodontitis patients with CVD in terms of changes in systemic inflammatory blood marker were subject to several methodological biases, mostly related to the inadequate follow-up periods of the included studies [51,54,55] or biasing approaches during the review process (analysis of studies providing low-quality data, with different designs, or including CVD patients of different risks) [51,55,58]. The reported biases prevent comparisons between reviews and the formulation of a firm conclusion related to the analyzed subject.

A non-significant decreasing trend of CRP values at 6 months after treatment for periodontitis patients with comorbid CVD was reported [46,59]. The greatest reductions were obtained for baseline CRP levels > 3 mg/L [60]. On the other hand, infection control through subgingival mechanical instrumentation reduced serum CRP concentrations by 0.69 mg/L (95% confidence interval: −0.97 to −0.40) after 6 months in systemically healthy individuals [60], to a degree equivalent to that obtained through traditional lifestyle modifications [61] or medication [62]. This information is of extreme importance since the reduction of CRP in CVD patients to less than 2 mg/mL decreased the risk of cardiovascular death by one-third independent of other risk factors [8].

Limited data are available regarding changes in serum calprotectin (S100A8/A9) levels among patients with periodontitis and CVD. A single study reported a notable rise in calprotectin levels in individuals with both periodontitis and coronary artery disease, which subsequently decreased 180 days after nonsurgical periodontal treatment alongside improvements in cardiac parameters [63].

While not statistically significant, a decrease in white blood cell (WBC) count was observed three months after nonsurgical periodontitis treatment in patients with CVD [64]. On the other hand, significant reductions in CRP and WBC count, along with improvements in periodontitis-related parameters, were reported one month after nonsurgical periodontal treatment in patients with both periodontitis and CVD [50]. Notably, a more pronounced reduction in WBC count was noted in patients with CVD compared to the control group [50].

### 2.3. Adverse Short-Term Effects of Subgingival Mechanical Instrumentation in Periodontitis Patients

Subgingival decontamination via conventional mechanical instrumentation (quadrant-based) typically occurs over multiple sessions spanning several weeks, with each session focusing on scaling a specific quadrant of the mouth.

An alternative therapeutical protocol known as full-mouth scaling (FMS) proposes treating the entire mouth within 24 h across one or two sessions, to mitigate the risk of recontamination. This approach aims to prevent the re-infection of recently cleaned periodontal pockets from untreated areas. Another option, full-mouth disinfection (FMD), combines FMS with extensive oral disinfection using chlorhexidine. In certain clinical situations, both quadrant-based and full-mouth scaling approaches may be associated with adjuvant antimicrobials applied locally or administrated systemically [42].

However, the additional local benefits of FMS and FMD compared to quadrant instrumentation are minimal, and clear recommendations for their use, particularly for FMD, are lacking [42,65,66]. While the FMS approach may seem logical to prevent recontamination, its implementation in clinical practice may present challenges, particularly in cases of moderate or severe generalized periodontitis (stage II or III periodontitis) where significant post-therapeutic bacteremia could occur.

The most recent systematic review [65] assessed the effects of full-mouth decontamination protocols in comparison to the conventional quadrant approach in periodontitis patients, highlighting an increase in body temperature as a notable short-term adverse event following full-mouth treatments [65]. This elevation in temperature may stem from repeated transient bacteremia occurring during subsequent subgingival instrumentation sessions [65].

Evidence suggests that full-mouth protocols can trigger acute systemic reactions within 24 h, manifested by a three-fold rise in CRP, a two-fold increase in IL-6, and a slight elevation in tumor necrosis factor-alpha levels in the full-mouth instrumentation group compared to the conventional quadrant instrumented group [67]. An acute-phase response one day after one-stage full-mouth instrumentation versus the quadrant approach was also reported in subjects with comorbid type 2 diabetes [68]. Consequently, there is a recommendation to carefully weigh the choice of therapeutic approach, considering the patient’s overall health status [42]. The European Federation of Periodontology and the World Heart Federation experts advise that in periodontitis, for individuals without extended edentulism, with different severities of CVD, and regardless of specific medications, nonsurgical periodontal therapy should preferably be delivered over several 30 to 45 min sessions to minimize acute systemic inflammation [16]. Moreover, individuals at risk of endocarditis ought to receive antibiotic prophylaxis in accordance with the current recommendations [16].

Other short-term effects (within 24 h) of full-mouth decontaminating approaches included tooth discoloration and alterations in taste perception among participants in the FMD group [65], resulting in challenges regarding participants’ adherence to the study during the 60-day follow-up period [69]. Furthermore, undesirable outcomes subsequent to full-mouth instrumentation in contrast to the quadrant treatment encompassed heightened analgesic usage and the occurrence of oral herpes in the full-mouth treatment groups [70]. Conversely, several randomized controlled trials revealed no discernible disparities in adverse reactions between groups subjected to full-mouth approaches versus those undergoing conventional instrumentation [71,72].

In order to more comprehensively examine the systemic implications of infection control in individuals with both periodontitis and CVD, it is recommended that future RCTs incorporate larger groups of participants, including those with associated pathologies, implement more rigorous control over confounding variables, and explicitly address outcomes related to both periodontal health and cardiovascular health [40].

## 3. Discussions

Until now, the evidence has been inconclusive regarding whether periodontal therapy can aid in preventing CVD in individuals diagnosed with periodontitis, or whether long-term management of periodontal inflammation could improve systemic health [40]. Furthermore, it remains unclear whether nonsurgical periodontal treatment combined with antibiotics can reduce all-cause mortality at the 12-month follow-up compared to supragingival scaling alone in patients with CVD [40].

CRP and S100A8 were chosen as relevant markers of systemic inflammation associated with both periodontitis and CVD. Increased circulating levels of CRP have been reported in periodontitis [73,74,75] as well as in CVD patients [61,76]. Subclinical inflammation, as indicated by plasma CRP levels, was associated not only with the presence of coronary atherosclerosis but also with the severity of coronary plaque burden and the clinical manifestation of ischemia [77]. Furthermore, even a slight yet persistent increase in CRP or high-sensitivity CRP (hs-CRP) has been correlated with the presence of coronary artery disease and a heightened risk of future cardiovascular events among apparently healthy individuals [78]. Considering the systemic risks associated with increased serum CRP levels [77,78] and the observed decrease in CRP concentrations following nonsurgical therapy in periodontitis patients with concurrent CVD [43,46,60], immediate infection control in these individuals is imperative.

S100A8/9, also known as calprotectin, belongs to the broader family of S100 calcium-binding proteins and has the ability to bind zinc. These proteins are predominantly found in neutrophils, monocytes, or macrophages [79] and contribute to the production of proinflammatory mediators. However, under certain specific conditions, they can also demonstrate anti-inflammatory effects [80]. Elevated serum [81] and salivary levels [82,83] of S100A8/A9 were associated with periodontitis patients. Increased concentrations of salivary S100A8 were associated with elevated metalloproteinase-9 levels, which supports the possible use of these salivary biomarkers as a rapid test kit for the early detection of periodontitis [83]. Elevated release of S100A8/A9 appears to exert deleterious effects during myocardial infarction and is linked to an unfavorable long-term prognosis [84], while the inhibition of S100A8/A9 has been effective in myocardial infarction models [80]. The reduction of serum S100A8/A9 through nonsurgical periodontal therapy [63] seems an appealing perspective in CVD individuals. However, more clinical trials should further establish the impact of periodontitis infection control on S100A8/A9 concentrations in this group of patients.

Cellular circulating parameters are reliable hematological surrogates for monitoring body inflammation [85]. Periodontitis has been associated with hematological changes [86,87,88], mostly related to a higher white blood cell (WBC) count [86]. An increased WBC count is a marker of systemic inflammation in CVD [89] and a risk factor for coronary heart disease, ischemic stroke incidence, and CVD mortality in Americans [90]. Infection control through subgingival mechanical instrumentation in periodontitis patients has been linked to a decrease in WBC count, which could confirm the possible causal link between periodontitis and leukocyte count [86]. However, there are inconsistent findings regarding the decrease in WBC count following nonsurgical therapy in individuals with both periodontitis and CVD [50,64], which signals a need for further insights into this matter.

## 4. Material and Methods

### 4.1. Literature Search

A literature search was conducted in the electronic databases PubMed MEDLINE and Cochrane Library, without any restriction regarding the publication date, with the last search being performed in February 2024. The following index terms and Boolean operators were used in the established search strategy: “Cardiovascular Disease” AND “Periodontal Treatment”; “Cardiovascular Disease” AND “Periodontal Therapy”; “Inflammatory Markers” AND “Periodontal Treatment” AND “Cardiovascular Disease”; and “Inflammatory Markers” AND “Periodontal Therapy” AND “Cardiovascular Disease”; “Blood Cell Count” AND “Periodontal Treatment” AND “Cardiovascular Disease”; “Blood Cell Count” AND “Periodontal Therapy” AND “Cardiovascular Disease”.

### 4.2. Inclusion Criteria

We included randomized controlled trials, clinical controlled trials, and systematic reviews with or without meta-analysis evaluating the effects of nonsurgical periodontal therapies (subgingival mechanical instrumentation with or without adjunctive antimicrobials) [42] in individuals diagnosed with periodontitis, with or without CVD. The following therapeutic outcomes were considered: cardiovascular-related events (all-cause and CVD-related death, angina, myocardial infarction, stroke), blood cell counts, systemic inflammation parameters, and local and systemic adverse events induced by periodontal therapy. Non-English-language articles, experimental studies on animals, and clinical studies that assessed the effect of periodontal treatment on cardiovascular risk in patients with other systemic comorbidities were excluded from the search.

### 4.3. Data Collection and Synthesis

Screening of the identified papers was performed independently by two authors (I.C.M. and A.S.) by checking the titles, abstracts, and, finally, full texts of the retrieved papers regarding the predefined eligibility criteria. Screening of the reference lists of selected papers was also carried out, and the identified papers underwent the check of eligibility (I.C.M., A.P.). Publications with an unclear methodology but considered relevant and cited by relevant reviews were included in the full-text assessment to avoid missing potentially relevant information. In the event of ambiguity, consensus through discussion between the three authors participating in the literature search was achieved.

The identified clinical studies and systematic reviews were organized in three main categories regarding the influence of nonsurgical periodontal therapies on cardiovascular-related events, variations in systemic inflammatory blood markers and blood cell counts induced by the subgingival mechanical instrumentation, as well as the main adverse events associated with the subgingival mechanical instrumentation.

## 5. Conclusions

Periodontitis has the capacity to initiate and worsen the progression of CVD and its acute complications through intricate pathogenetic mechanisms, largely involving the release of proinflammatory molecules.

In patients with both periodontitis and concurrent CVD, subgingival mechanical instrumentation can mitigate systemic inflammation, as evidenced by reduced levels of proinflammatory markers such as C-reactive protein, calprotectin, and white blood cell counts. Decreasing the systemic inflammatory burden lowers the risk of developing CVD and experiencing acute events.

To minimize acute systemic inflammatory effects, it is advisable to implement quadrant-based instrumentation protocols at staggered intervals rather than using full-mouth subgingival mechanical instrumentation in individuals with periodontitis and CVD.

## Figures and Tables

**Figure 1 antibiotics-13-00359-f001:**
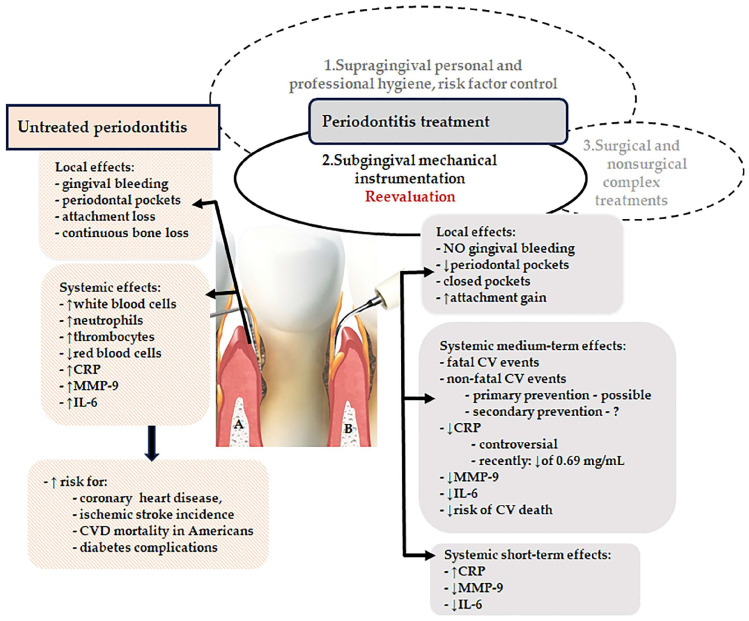
Synoptic view of local and systemic effects in untreated and treated periodontitis. (A) Contaminated periodontal pocket, (B) Cleaned pocket through subgingival mechanical instrumentation (CVD, cardiovascular disease, CRP, C-reactive protein, IL-6, interleukin-6, MMP-9, metalloproteinase-9; NO, nitric oxide), ↓ decrease, ↑ increase, [Photo—Free for Canva Pro].

## Data Availability

Not applicable.

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
