# Peer review of "Systemic Impact of Subgingival Infection Control in Periodontitis Patients with Cardiovascular Disease: A Narrative Review"

_antibiotics, 2024, doi:10.3390/antibiotics13040359_

Round 1

Reviewer 1 Report

Comments and Suggestions for Authors

Thank you for the opotunity to review this review of Periodontitis and CVD.

I am missing the use of the actual perio classification with the different stages of periodontitis and its importance on CVD adverse effects when citing the presented studies.

Comments on the Quality of English Language

Much too long english sentences. The whole review should be reduced to in maximum 5 pages. The table is missleading while presenting commercal producht not longer available in europe.

Author Response

Dear Sir,

We appreciate your insightful suggestion and the time you dedicated to reviewing our manuscript. In the attachment, you will find our responses to your suggestions on the platform, followed by detailed point-by-point responses to each of your comments in the PDF version.

Reviewer 2 Report

Comments and Suggestions for Authors

Some modifications are required

(The Authors must see my remarks)

Author Response

Dear Sir,

We appreciate your insightful suggestion and the time you dedicated to reviewing our manuscript. In the attachment, you will find our responses to your suggestions on the platform, followed by detailed point-by-point responses to each of your comments in the PDF version.

As part of this revision, we had to remove detailed explanations, some paragraphs, and some specific comments that you highlighted in your review. These sections were omitted to streamline the focus and maintain the necessary brevity, as per the guidelines for submission. We  removed the table in which were presented the commercial products as a request of a reviewer.

In conclusion, it's important to note that in response to the requests of the other two reviewers, particularly the third reviewer who insisted on organizing the article into conventional sections, significant modifications have been made to the manuscript.

We believe the changes made reflect the core objectives of the study while adhering to the journal's requirements for conciseness and clarity.

Reviewer 3 Report

Comments and Suggestions for Authors

Thank you for submitting this manuscript. I would like to suggest some points to the authors:

Even though it is a review, the article must contain the same sections: introduction, materials and methods, results, discussions, and conclusions.

Materials and Methods: How many studies were included in the review: what were the inclusion/exclusion criteria? From which databases were the studies selected? How were the studies grouped?

Results: They must be related to the study's purpose. For each group established in the materials and methods, it must be specified how many studies were included, what was aimed at, and what conclusions were reached.

The conclusions are too general; they must be related to the study's purpose and the results obtained.

The abstract has the same structure as the article: introduction, materials and methods, results, discussions, and conclusions.

Author Response

Dear Sir,

We appreciate your insightful suggestion and the time you dedicated to reviewing our manuscript.

As part of this revision, we had to remove detailed explanations, some paragraphs, and table 1 as requests of the other two reviewers. In conclusion, it's important to note that significant modifications have been made to the manuscript due to the three reviewers’ comments.

We have taken your comments into serious consideration and are pleased to inform you that we have comprehensively revised the manuscript. The changes span across the structure of the Abstract, Introduction, Materials and Methods, Results, Discussions, and Conclusions sections. 

In the attachment please find the full answer to your requests.

Round 2

Reviewer 1 Report

Comments and Suggestions for Authors

Thank you for the second chance to review your amnuscipt. CVD is a widespread health problem in elderly and geriatric patients. Adjunctive antibiotic treatment metrondiazole (alone as discussed in the EFP Paper by Herrera 2022) or azithromycin (alone as presented the first time by Gomi et al 2007) starting 1-2 days before AIT is still not discussed. It is quite difficult to treat geriatric patients 4 times. The aspects of endocarditis risc and prophyaxis should be presented at least with one sentence in the conclusions.

Author Response

Dear Sir,

               We would like to extend our deepest gratitude for the time and effort you've invested in reviewing our medical narrative review and thank you for your valuable feedback on our manuscript. Based on your suggestion, we have carefully revised the article and we added the information’s regarding prophylaxis of endocarditis in section Results, rows 270-271, as you recommended.

               Thank you once again for your thorough review and for helping us improve the quality of our manuscript.

Reviewer 3 Report

Comments and Suggestions for Authors

Dear Authors,
Thank you for submitting this manuscript. I would like to suggest some points to the authors:
Results: 26 articles were evaluated, not 23 (line 148)
PRISMA flowchart should be entered.

Author Response

Dear Sir,

               We sincerely thank you for your insightful observations and deeply apologize for the oversight in our calculations. Your feedback is invaluable to us, and we are committed to ensuring the integrity and accuracy of our research. Upon receiving your comments, we immediately undertook a thorough re-examination of our narrative review. We have reevaluated the number of articles considered in our narrative review and have updated the results section with the correct data accordingly. Taking your suggestion to incorporate a PRISMA flowchart into consideration, please note that our article was originally designed as a narrative review from the outset. We appreciate your patience and understanding as we corrected this inadvertent error (section Results, rows 149-150). It is our hope that these revision adequately address your concerns and strengthen the contribution of our narrative review to the field.

               Thank you once again for your thorough review and for helping us improve the quality of our manuscript.